# Characterization of a Novel *Thermobifida fusca* Bacteriophage P318

**DOI:** 10.3390/v11111042

**Published:** 2019-11-08

**Authors:** Jatuporn Cheepudom, Tzu-Ling Lin, Cheng-Cheng Lee, Menghsiao Meng

**Affiliations:** Graduate Institute of Biotechnology, National Chung Hsing University, Taichung City 40227, Taiwan; cheepudomj@gmail.com (J.C.); mljj8912@gmail.com (T.-L.L.); lichanjan@gmail.com (C.-C.L.)

**Keywords:** thermostable bacteriophage, *Thermobifida fusca*, viral genome packaging, *Siphoviridae*

## Abstract

*Thermobifida fusca* is of biotechnological interest due to its ability to produce an array of plant cell wall hydrolytic enzymes. Nonetheless, only one *T. fusca* bacteriophage with genome information has been reported to date. This study was aimed at discovering more relevant bacteriophages to expand the existing knowledge of phage diversity for this host species. With this end in view, a thermostable *T. fusca* bacteriophage P318, which belongs to the *Siphoviridae* family, was isolated and characterized. P318 has a double-stranded DNA genome of 48,045 base pairs with 3′-extended COS ends, on which 52 putative ORFs are organized into clusters responsible for the order of genome replication, virion morphogenesis, and the regulation of the lytic/lysogenic cycle. In comparison with *T. fusca* and the previously discovered bacteriophage P1312, P318 has a much lower G+C content in its genome except at the region encompassing ORF42, which produced a protein with unknown function. P1312 and P318 share very few similarities in their genomes except for the regions encompassing ORF42 of P318 and ORF51 of P1312 that are homologous. Thus, acquisition of ORF42 by lateral gene transfer might be an important step in the evolution of P318.

## 1. Introduction

Lignocellulosic wastes produced within the agriculture and forestry sectors can provide myriads of renewable carbon resources for the production of biofuels and various green chemicals. It could prove useful if the recalcitrant structure of lignocellulose can be broken and the fermentable sugars inside can be retrieved in economically feasible ways. In nature, some bacteria, e.g., *Clostridium thermocellum* [1], and fungi, e.g., *Trichoderma reesei* [2], have evolved complex enzymatic systems to degrade cellulose. Therefore, various types of cellulase from such cellulolytic microorganisms have long been utilized in attempts to release sugars from cellulosic biomass. Among various cellulolytic microbes, *Thermobifida fusca* has gained considerable attention in recent years due to its great physiological and cellulolytic characteristics.

*T. fusca* is an aerobic, moderately thermophilic, soil bacterium belonging to Actinobacteria. It is known as an excellent producer of cellulolytic enzymes including various types of cellulase, xylanase, and many other glycoside hydrolases [3,4] as well as lignin degradation-related oxidoreductase [5,6]. Whole genome sequencing of *T. fusca* YX, a type strain of the species, was completed in 2007 [7], providing information regarding the complexity and regulation of cellulolytic enzymes. From an application perspective, *T. fusca* can not only be a provider for various thermostable enzymes with industrial potential, but also an alternative host for metabolic engineering, by which lignocellulosic wastes can be transformed into value-added chemicals. For example, an engineered *T. fusca* strain that carries an exogenous gene of bifunctional butyraldehyde/alcohol dehydrogenase was able to convert untreated lignocellulosic biomass to 1-propanol [8]. Another engineered strain whose chromosome carried the pyruvate carboxylase gene from *Corynebacterium glutamicum* could produce malic acid based on the consumption of corn stover [9]. Despite these successful pioneering results, there still remains a high demand for more molecular biology tools and reagents to maximally fulfill the application potential of *T. fusca* in the biotransformation of lignocellulosic wastes into valuable chemicals.

Recently, the recombination system of temperate bacteriophages was exploited to develop many recombineering tools. For examples, the bacteriophage λ Red system, comprising *exo* (*α*), *beta* (*β*), and *gam* (*γ*), has been successfully demonstrated in genome editing of different prokaryotic cells since its first introduction in 2000 [10]. The Cre/*loxP* system from phage P1 represents another achievement, which has been widely used in editing bacterial, plant, and mammalian genomes [11]. Given the ubiquitous nature of phages, seeking new phages in diverse environments may lead to the discovery of novel genes and proteins with application potential in medicine, agriculture, biotechnology, bioremediation, energy generation, and others.

To date, over 1930 bacteriophage genomes have been deposited in the databases of the National Center for Biotechnology Information (NCBI). Nonetheless, only one bacteriophage that infects *T. fusca* has been documented [12]. This bacteriophage, named P1312, was classified in the *Siphoviridae* family because it has an icosahedral head and a long flexible non-contractile tail. P1312 contains a dsDNA genome of 60,284 bp with circularly permuted ends. P1312 is extremely thermostable; incubation at 90 °C for 45 min did not affect its infectivity. Given the application potential of *T. fusca* in the biotransformation of lignocellulosic biomass, we sought out more bacteriophages that could infect *T. fusca*. A novel bacteriophage designated P318 was isolated in this study. Infectivity and genome characteristics of P318 are reported herein.

## 2. Materials and Methods

### 2.1. Isolation and Purification of P318

A number of fertile soil samples, collected from local flower shops, were screened for the presence of bacteriophages that infect *T. fusca* according to the protocol described previously [12]. Briefly, one ml of the overnight culture of each soil sample was irradiated for 30 s under ultraviolet light (120 mJ/cm^2^) and immediately mixed with 10^7^
*T. fusca* spores and 15 mL CYC soft agar medium at 50 °C. The mixture was then poured onto a regular CYC agar plate and incubated at 50 °C until the appearance of plaques was visible on this double-layer agar plate.

Several isolated phage plaques were selected from a double-layer agar plate and each was transferred into 2 mL *T. fusca* culture. After 2 days incubation at 50 °C, the culture was centrifuged at 12,000 g, 4 °C, for 10 min. The resulting supernatant was taken and mixed with 1.5 M NaCl and 10% (*w/v*) polyethylene glycol (PEG) 8000. After sitting on ice for 1 h, the mixture was centrifuged at 12,000 g, 4 °C, for 15 min to precipitate the phage particles. These pelleted virions were then subjected to protein analysis by standard SDS-polyacrylamide gel electrophoresis (PAGE) and detection by Coomassie blue staining. In addition, the original pellet obtained from the 500 mL culture broth was suspended in 10 mL TBS buffer (50 mM Tris-HCl (pH 8.0), 150 mM NaCl) and treated with 0.25 U/mL DNase I and 10 µg/mL RNase A (Takara Bio, Kusatsu, Japan), at 30 °C for 1 h. This solution was passed through 0.45 µm pore size filter and then loaded into a Tricorn 10/600 column packed with Sephacryl S-500 HR (Amersham Biosciences, Little Chalfont, UK). Gel filtration chromatography was performed with TBS buffer at a flow rate of 0.7 mL/min. The virions were eluted in the void volume of the column, as shown in the first OD_280_ absorption peak. The morphology of the purified phage particles was observed under the transmission electron microscope Jeol LEM-1400 with 120 kV accelerating voltage as described previously [12].

### 2.2. One Step Growth Cycle of P318

The overnight culture of *T. fusca* was mixed with P318 at a multiplicity of infection (MOI) of 0.001. After incubation for 10 min at room temperature to allow for phage attachment, the mixture was centrifuged at 12,000× *g*, 4 °C, for 5 min. The pelleted cells were washed twice with CYC medium, and suspended in 50 mL CYC medium and then incubated at 50 °C. Aliquots of the culture were taken at indicated intervals and the phage titers in the clarified samples were then determined by the double-layer plaque assay.

### 2.3. Phylogenetic Analysis

Amino acid sequences of the terminase large subunit from a variety of tailed-bacteriophages were retrieved from the NCBI protein database and aligned using the ClustalW with default parameters in MEGA 6.0 [13]. Phylogenetic tree was built by the neighbor-joining method and phylogenies were determined by bootstrap analysis of 1000 replicates.

### 2.4. Genome Organization of P318

Whole genome sequencing of P318 was performed using Illumina Miseq (Tri-I Biotech, Inc., Taiwan). The count of reads was 115,622 with the average length of 175 bases per read. The sequence data could be assembled into a single contig of 48,045 bp using the de novo assembly algorithm of CLC Genomics Workbench v 9.5 (Qiagen, Aarhus, Denmark). The nucleotide sequence of the genome of P318 is deposited in the GenBank under accession number MK240575. Identification of potential open reading frames (ORFs) within the phage genome was performed by using the Bacterial Annotation System [14], ORF finder [15], and GeneMark [16]. The RBS calculator [17] was further used to confirm the presence of the ribosome binding site before the predicted ORFs. Annotation of the protein encoded by each ORF was primarily based on the BLASTp-PSI program, NCBI.

### 2.5. COS Site of P318

The genomic DNA of P318 was extracted and purified using the standard phenol-chloroform extraction method. To determine the distal ends of P318, 3 µg of genomic DNA was digested with BAL-31 exonuclease (abbreviated hereafter as BAL-31) at 30 °C for different intervals. The digestion reaction was stopped by incubation with 20 mM EGTA at 65 °C for 15 min. The DNA in all samples was purified again by the phenol-chloroform method, followed by a fast digestion with *Xho*I or *Hind*III at 37 °C for 5 min. The restriction patterns of the double digested DNA were analyzed by electrophoresis using a 0.8% agarose gel.

One of the *Hin*dIII-digested distal fragments was recovered from the agarose gel and purified by the Wizard SV gel clean up kit (Promega, Madison, WI, USA). The DNA fragment was then treated with T4 DNA polymerase at 12 °C for 20 min, and the resulting blunt fragment was inserted into pUC19. The sequence of the inserted fragment in pUC19 was determined by the Sanger dideoxynucleotide sequencing method using M13 forward sequencing primer (5′-GTTTTCCCAGTCACGAC-3′) and reverse primer (5′-ACAGGAAACAGCTATGAC-3′). The actual nucleotides at one end of the P318 genome was sequenced using the primer cosF’ (5′-ACTACCTAGGAAGGTGAAG-3′), which was designed based on the information obtained from the above pUC19 derivative.

### 2.6. Identification of Proteins Associated with P318 Virions

The protein composition of purified virions of P318 was identified by tandem mass spectrometry using an Applied Biosystems QStar LC-MS/MS spectrometer (Life Technologies, Carlsbad, CA, USA). The obtained spectrometry information was analyzed with the Mascot software (Matrix Sciences, London, UK) using the GenBank non-redundant protein database and the database created specifically in this study based on the predicted open reading frames of P318 (Table 1). The parameter settings for Mascot analysis were as follows: mass values, monoisotopic; protein mass, unrestricted; peptide mass tolerance, ±0.5 Dalton; fragment mass tolerance, ±0.5 Dalton; and maximal missed cleavages, 2.

## 3. Results and Discussion

### 3.1. Isolation and Purification of P318

P1312 was isolated recently and represents the first well-documented phage that infects *T. fusca*. To learn more about the diversity of *T. fusca* bacteriophages, more soil samples were collected and screened for phages. One of the samples, which contained chicken manure compost, gave rise to lytic plaques on the lawn of *T. fusca* (Figure 1A). Several isolated plaques were randomly selected and each was inoculated into *T. fusca* broth. Phage particles in the culture medium were collected by the NaCl/PEG 8000 precipitation method and subsequently subjected to SDS-PAGE analysis. All the selected plaques resulted in almost identical protein banding patterns but distinct from that of P1312 (Figure 1B), suggesting that the isolated plaques resulted from a novel *T. fusca* phage (referred hereafter as P318). To study P318, the pelleted virions collected from 500 mL culture broth by the NaCl/PEG 8000 method were suspended in 10 mL buffer, which contained DNase I and RNase A to eliminate nucleic acids apart from the virions. The mixture was then passed through a Sephacryl S-500 HR column (Figure 1C) to separate the virions from other substances in the mixture. The first peak of the elution profile represented the purified virions.

The nucleic acids enveloped in the purified virions were extracted with the phenol/chloroform method and subjected to enzymatic digestion with BamHI or SacI. Digestion of the P318 genome with BamHI generated 12 restriction bands, whereas digestion with SacI resulted in 7 restriction bands (Figure 1D). The DNA restriction patterns of P318 were different from those of P1312, confirming the distinction of P318 from the previously discovered P1312.

### 3.2. Morphology and Infectivity of P318

The morphology of P318 was examined by transmission electron microscopy. The P318 virion has a hexagonal head that is 40 nm in diameter and a long non-contractile tail that is 285 nm in length and 8 nm in diameter (Figure 2A). Therefore, P318 belongs to the *Siphoviridae* family. In terms of the appearance, P318 is slightly different from P1312 that has a head that is 56 nm in diameter and a tail that is 250 nm in length.

Metal ions are usually involved in adsorption and penetration of a virion into its host. Calcium ions are particularly important because they may enrich the local concentration of virion at the host surface or increase the accessibility of the cell surface receptors to the virion by altering the receptor structure [18]. The effect of CaCl_2_ on P318 infectivity was examined by adding CaCl_2_ at different concentrations into the soft CYC agar in the double-layer plaque assay. Addition of 0.1 mM CaCl_2_ was optimal, resulting in 70% more plaques being gained in comparison to the control without CaCl_2_ addition. This result is different from the previous observation on P1312 that actually showed infectivity was inhibited by addition of CaCl_2_ in the soft agar.

The thermal stability of P318 was examined by incubation of the virions at 60−95 °C for different time intervals, followed by the double-layer plaque assay (Figure 2B). Incubation at temperature ≤80 °C for up to 45 min did not inactivate the infectivity of P318. Nonetheless, incubation at 90 °C started to inactivate the phage infectivity. P318 was inferior to P1312 in this regard, because P1312 was resistant to a 45-min incubation period at 90 °C [12].

The infectivity of P318 in terms of burst size and latent period was also characterized. The burst size was calculated based on the one-step growth curve of P318 with MOI of 0.001 at 50 °C (Figure 2C). Approximately 92% of virions already adsorbed to the host after a 10-min incubation period. The titer of virions released from the infected cells into the culture medium at different time intervals was determined, and this resulted in a triphasic one-step growth curve. Based on the curve, P318 has a short 10 min latent period with the burst size of 60 pfu per infected cell. This value was close to 57 pfu per infected cell in the case of P1312 [12].

### 3.3. Genome Structure of P318

Analysis of the next-generation sequencing (NGS) data suggested that P318 has a circular genome of 48,045 bp with 54.6% G+C content. In comparison with 67.5% G+C content of *T. fusca*, the G+C content of P318 genome was unexpectedly low. Although a circular genome was suggested, P318 should have a linear genome since all the tailed-bacteriophages contain single linear dsDNA [19]. The type of the genomic ends of a tailed-bacteriophage depends on how a unit-length genome is cut from the concatemeric genome and packaged into the procapsid [20]. The amino acid sequence of the terminase large subunit has been used to predict the type of genome ends. The termini of a phage genome are classified into 5 different types: (1) single-stranded cohesive ends, which further split into either 5′ or 3′ cohesive end; (2) circularly permuted direct terminal repeats; (3) direct terminal repeats with no circular permutation; (4) long, non-permuted, direct terminal repeats; and (5) terminal host DNA sequences. Thus, the packaging type of P318 was predicted based on the alignment of its terminase large subunit with others whose genome termini have been determined. The resulting phylogenetic tree suggested that the genome of P318 has 3′-extended COS ends (Figure 3A). By contrast, the DNA packaging of P1312 was classified as the P22-like headful mechanism (Figure 3B).

To determine the distal restriction fragments of the P318 genome, the genomic DNA was first digested by BAL-31, which degrades both 3′ and 5′ termini of duplex DNA, at 30 °C for different incubation periods (0, 20, 40, and 60 min) and then by *Xho*I or *Hind*III for 5 min (Figure 4A,C). The restriction fragments that became shorter over the incubation time of BAL-31 would be the ones located at the ends of the genome. The restriction patterns of P318 after *Xho*I digestion is shown in Figure 4A. The fragment labeled as A was shortened in response to the pretreatment of BAL-31; accordingly, it should represent one of the distal fragments in the linear genome of P318. The other distal fragment, labeled as B, is indistinguishable in this gel because its size is close to the sizes of the neighboring bands. The rest of the fragments were labeled in numerical order from the largest to the smallest sizes. All the corresponding fragments are indicated in the genomic map of P318, which was drawn intentionally in a circular form, in Figure 4B. Similarly, the restriction patterns of P318 after *Hind*III digestion is shown in Figure 4C. The fragments labeled as X and Y became shorter in response to the pretreatment of BAL-31. The rest of the fragments were also numbered from the largest to the smallest sizes and indicated correspondingly in the circular map of P318 in Figure 4D. According to the mapping results, the termini of the phage genome should be located within the A+B fragment in Figure 4B and the X+Y fragment in Figure 4D. To determine the exact position of the Y fragment in the circular map, it was recovered from the agarose gel, blunted by T4 DNA polymerase, and inserted into pUC19 plasmid via *Sma*I restriction site. The recombinant plasmid was then sequenced using universal M13 primers. Based on the sequencing result, the Y fragment would be located downstream of the X fragment if the genome were circular as shown in Figure 4D. In other words, the actual terminus of P318 genome should be immediately upstream of the Y fragment. The primer cosF’, which anneals to the rear region of the X fragment and heads downstream (indicated in Figure 4D), was used in the sequencing reaction by using P318 genome as the template. As shown in Figure 4E, the sequencing reading was suddenly terminated at the nucleotide corresponding to the 33,630 bp in the circular map (Figure 4E). Thus, the gap from the terminated point to the end point of the Y fragment should represent the COS sequence. The Phageterm program was developed recently to predict the ends of a phage genome by using the NGS data of the analyzed phage [21]. This program relies on the detection of biases in the number of reads in NGS data. The Phageterm program predicted that P318 has a linear DNA with 3′ cohesive ends and the cohesive overhang has nucleotide sequence as CGCCGGGGGCC, which is in agreement with the laboratory data described above.

### 3.4. Putative Proteins Encoded by P318

At least 52 ORFs were predicted on the genome of P318. The homology search of each of the protein sequences was conducted by the PSI-BLAST program. The general information of the putative proteins regarding their sizes, physiological functions, and most similar homologs are listed in Table 1. The first ORF was tentatively assigned to the one encoding a ssDNA-binding protein, because it is closest to one of the COS ends. Among the ORFs, 46 ORFs start with methionine and 6 with valine. The longest ORF encodes a protein of 1899 amino acid residues, while the shortest one encodes a peptide of 62 amino acids. Thirty-seven ORFs encode proteins containing known functional domains, and 15 ORFs encode hypothetical proteins that have been annotated in other bacteriophages or bacteria.

The virions purified via gel filtration chromatography were subjected to protein analysis by LC-MS/MS. Peptidyl fragments derived from 22 putative proteins were detected by the method (Table 1). They included 9 structural proteins, 7 hypothetical proteins, 3 nucleotide metabolic enzymes (thymidylate synthase, phosphoesterase, and ribonucleoside-triphosphate reductase), 2 proteins that contain domains of unknown function (DUF), and a peptidoglycan-binding domain-containing protein (Table 1). It is worth noting that neither RNase A nor DNase I was detected by LC-MS/MS, suggesting that both nucleases used prior to gel filtration chromatography were completely eliminated in the purification step. Therefore, the proteins detected should be those that constitute or adhere to the purified virions. The presence of the three nucleotide metabolic enzymes with virions is interesting. It implies that they either play an unidentified role in the invasion process of P318 or interact nonspecifically with the viral structural proteins.

The genome of P318 is organized approximately into three clusters based on the presumable functions of the encoded proteins (Figure 5). The first cluster consists of ORF1–15, encoding proteins primarily responsible for ribonucleotide synthesis and replication of the genomic DNA. ORF1 encodes a ssDNA-binding protein (SSB), which is thought to interact with the viral polymerase (product of ORF5), primase (product of ORF12), and helicase (product of ORF14) in virus DNA replication. The protein encoded by ORF3 contains an adenylation domain, which shares a similar sequence with RNA ligase 2 (Rnl2). In some bacteriophages, Rnl2 repairs broken RNA strands by counteracting a host defense of specific cleavages in tRNA molecules [22]. ORF4 encodes a FAD-dependent thymidylate synthase that catalyzes the reductive methylation of 2’-deoxyuridine-5’-monophosphate (dUMP) to 2’-deoxythymidine-5’-monophosphate (dTMP). ORF6 encodes ribonucleoside-triphosphate reductase that catalyzes the reduction of ribonucleotides to the corresponding deoxyribonucleotides. ORF8 encodes a putative σ factor that might be used to initiate transcription of some phage genes. The product of ORF9 contains a metallophosphatase domain, a signature of SbcD-like exonucleases. ORF13 directs the synthesis of a resolvase. The bacteriophage resolvase generally has two essential roles: (1) debranching DNA structures prior to packaging the viral genome into the head particles and (2) degrading the host DNA [23]. The product of ORF15 is supposed to be an exonuclease containing the RecB-like exonuclease domain. The product of ORF16 belongs to CBS (cystathionine β synthase). Its function in relation to P318 is unknown.

The second cluster, ORF17−37, encodes structural components or proteins related to the morphogenesis of P318 virion. ORF17 and 18 encode the terminase small and large subunits, respectively. Phage terminase is a hetero-oligomer protein composed of small and large subunits. The terminase small subunit helps to position the large subunit at the packaging initiation site. Meanwhile, the large subunit acts as an endonuclease that cuts the viral genome and a motor that translocates a unit-length viral genome into procapsids. The protein encoded by ORF19 is a phage portal protein, forming a hole or portal that permits DNA passage during packaging and ejection. ORF20 produces a scaffold protein, auxiliary proteins present during the virus assembly [24]. ORF21 encodes the viral capsid protein. With the portal protein (ORF19 product), the capsid protein assemble itself into an empty procapsids, in which the viral DNA is translocated through the portal structure. The product of ORE23 is a head-tail connecter protein that joins the head and tail at the last step of morphogenesis. The phage tail components are presumably encoded by ORF27, ORF31, ORF32, and ORF37. Products of ORF28 and ORF29 belong to phage tail assembly chaperone proteins (TAC) that coat the tape measure protein (TMP), the product of ORF30, to prevent the latter from forming unproductive complexes. TMP dictates the phage tail length by a constant of 0.15 nm per amino acid residues [25]. Accordingly, the average size of P318 tail is calculated to be 285 nm long, consistent with the morphology of P318 observed under the transmission electron microscope. The lytic gene module of P318 includes the N-acetylmuramoyl-L-alanine amidase (ORF35), the holin protein (ORF36), and a peptidoglycan (PG)-binding domain (ORF38). P318 amidase contains an enzymatic catalytic domain (ECD) at the N terminus and a PG-binding domain at the C-terminus. Holin, a small membrane protein, accumulates in the cytoplasmic membrane until it reaches a critical concentration, by the time the membrane is permeable enough to allow the phage endolysin (amidase in the case of P318) to pass through. The protein encoded by ORF38 also contains a PG-binding domain but lacks an ECD. ORF39 and ORF40 are partially complementary to each other. ORF39 encodes a protein with a DUF3987 domain. Although the function of the domain is unknown, the production of the protein was verified by LC-MS/MS. ORF40 encodes a putative methyltrasferase.

The third cluster is less defined in terms of the functionality of the proteins encoded in this region. ORF42 encodes a hypothetical protein homologous to the ORF51-encoded protein of P1312. Interestingly, these two proteins were detected by LC-MS/MS, suggesting that their presence were more than “hypothetical”. ORF44 encodes a recombinase, which might be linked intimately to the mechanisms of chromosome integration and excision. ORF45 and ORF46 presumably encode transcriptional regulators belonging to the MerR family. They may regulate the expression of certain genes in response to environmental stimuli, such as oxidative stress, heavy metals, or antibiotics [27]. According to the gene composition, P318 is likely a temperate phage. Confirmation of the lifestyle of P318 and identification of integration site on the host chromosome are interesting queries, deserving further studies.

### 3.5. Comparison between P318 and P1312 Genomes

Genomes of P1312 and P318 were compared in nucleotide sequences by Mauve alignment [28] (Figure 6A). Overall, the genomes of P318 and P1312 are quite dissimilar. Homologous sequences were only present at regions encoding the phage tail proteins and the ECD of endolysin as well as between ORF42 of P318 and ORF51 of P1312. The G+C content of the genome of P318 is 54.6%, which is much lower than the value of P1312 (65.9%). Distribution of G+C residues along the whole genome of P318 was further examined (Figure 6B). The region encompassing ORF42 had a much higher G+C content compared with the rest of the genome, suggesting that this region was incorporated into the genome of P318 by horizontal gene transfer perhaps in the latter phase of the evolution history of P318.

## 4. Conclusions

Until now P1312 and P318 are the only two reported *T. fusca* bacteriophages with genomic information. They both belong to the *Siphoviridae* family and have similar morphology except P318 has a smaller head and a longer tail. The smaller cavity in the head of P318 agrees with its need to accommodate a shorter genome. Both bacteriophages are thermostable, indicating that they have evolved to survive the high temperatures of the composting process. Still, P1312 is even more stable than P318 when they were heated at 90 °C. It would be interesting to determine whether or not the higher G+C content in the genome of P1312 relates to this difference. Considering the genome structure, P318 contains a genome of 48,045 base pairs with 3′-extended COS, while P1312 has a genome of 60,284 base pairs with circularly permuted ends. Overall, the genomes of P318 and P1312 are quite different in nucleotide sequence. However, it is noteworthy that the ORF42-encoded protein of P318 shares a significant similarity, 70%, to the ORF51-encoded protein of P1312. It is speculated that ORF42 of P318 and ORF51 of P1312 have the same evolutionary origin and these two hypothetical proteins may play an important role in the infection of the phages to *T. fusca*.

## Figures and Tables

**Figure 1 viruses-11-01042-f001:**
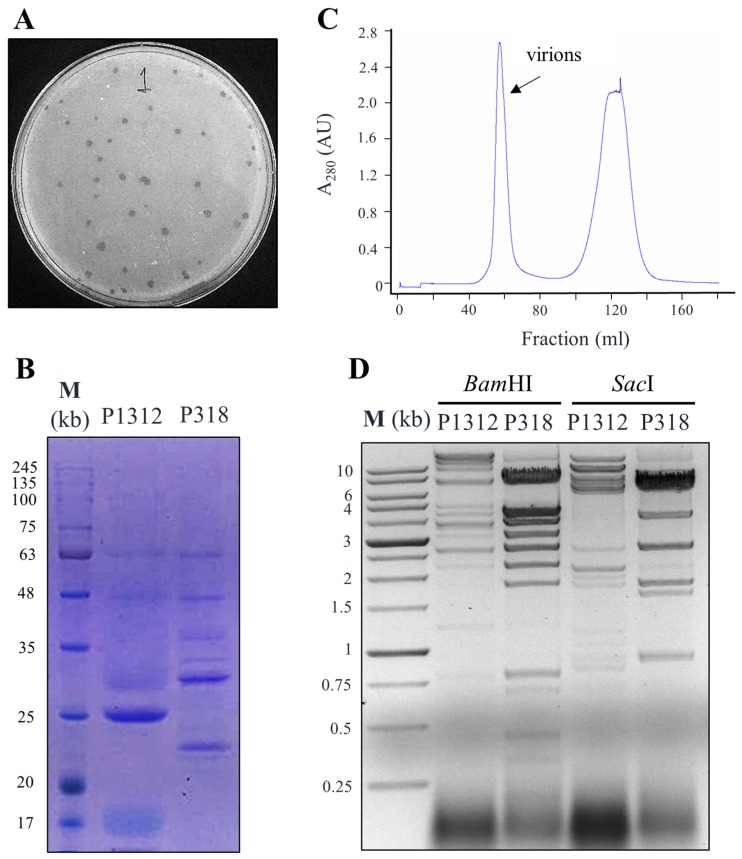
Isolation and purification of *T. fusca* bacteriophage P318. (**A**) Plaques formed in the lawn of *T. fusca* 10-1 strain in a CYC double-layer agar plate. (**B**) Protein compositions of P318 and P1312 virions analyzed by 12% polyacrylamide SDS-PAGE. (**C**) P318 virions were purified by the gel filtration chromatography using a Sephacyl S-500 column. (**D**) The differential restriction patterns between P318 and P1312 after digestion with *Bam*HI or *Sac*I restriction enzyme.

**Figure 2 viruses-11-01042-f002:**
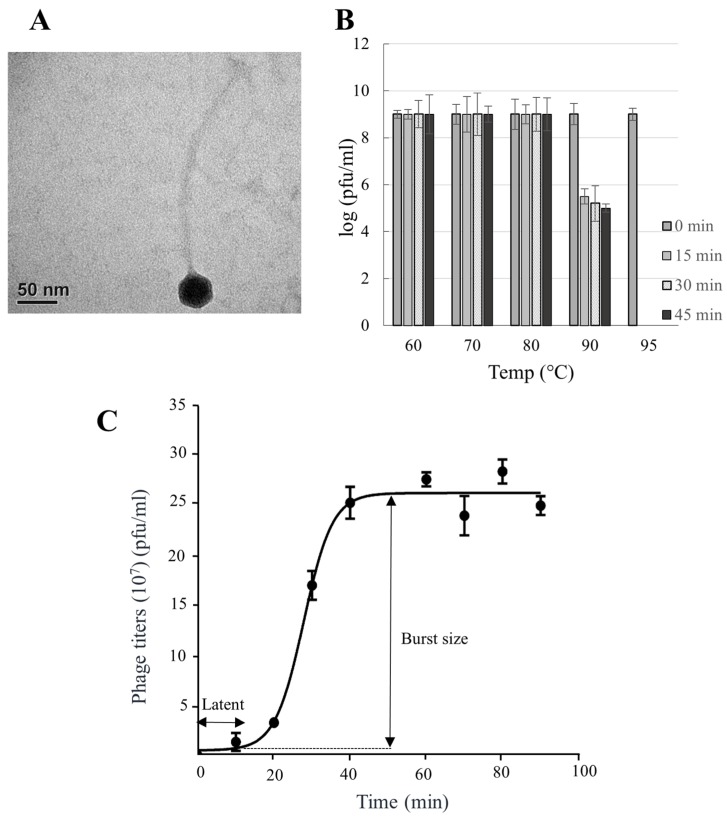
Morphology and infectivity of P318. (**A**) Virions of P318 observed under the transmission electron microscope at 400,000× magnification. (**B**) Thermal stability of P318 virions showing indicated temperatures at different time periods (0, 15, 30, and 45 min). The phage titer was determined by the double-layer agar plate assay. Values are the means of three determinations with ± SD. (**C**) One-step growth cycle of P318. Infection of *T. fusca* by P318 was initiated at the condition of 0.001 MOI at 50 °C. The latent period and burst size of P318 were determined from the growth curve. The error bar represents the standard deviation of three independent experiments.

**Figure 3 viruses-11-01042-f003:**
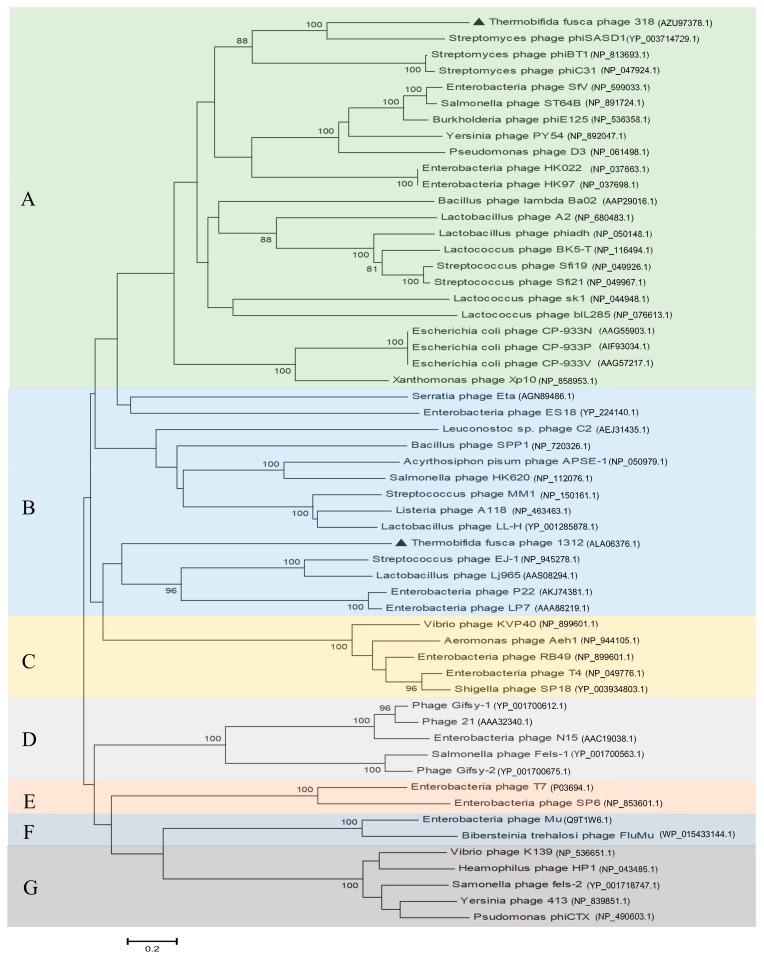
Phylogenetic analysis of the terminase large subunit of P318. The neighbor-joining phylogenetic tree shows the classification of the large terminase subunit of P318, P1312, and other phages with known packaging mechanisms. (**A**) 3′-extended COS ends; (**B**) P22-like headful; (**C**) T4-like headful; (**D**) λ like 5′-extended COS ends; (**E**) T7-like direct terminal repeats; (**F**) Mu-like headful; (**G**) P2-like 5′-extended COS ends. Bootstrap analysis of the phylogenetic tree was performed using 1000 repetitions.

**Figure 4 viruses-11-01042-f004:**
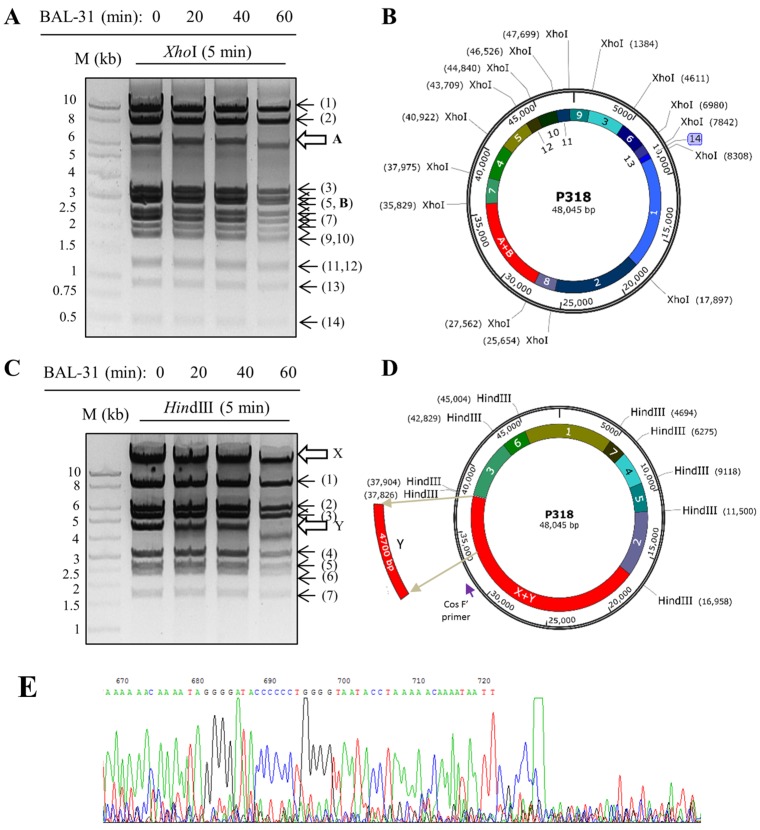
Determination of the COS ends of the P318 genome. (**A**) The genome was pre-digested with BAL-31 exonuclease for the indicated time periods and fast digested with *Xho*I for 5 min. The restriction patterns were analyzed by agarose gel electrophoresis and the fragments are labeled in numerical order according to their sizes except fragment A, which was shortened by the action of BAL-31. (**B**) The restriction fragments are positioned correspondingly in the circular map of the P318 genome. (**C**) The restriction patterns of *Hind*III were also generated as described above. The sizes of fragment X and Y were reduced by the action of BAL-31. (**D**) The restriction fragments of *Hind*III are positioned correspondingly in the circular map of the P318 genome. (**E**) The terminal sequence of the genome was determined by the Sanger method using cosF’ primer depicted in Figure 4D.

**Figure 5 viruses-11-01042-f005:**
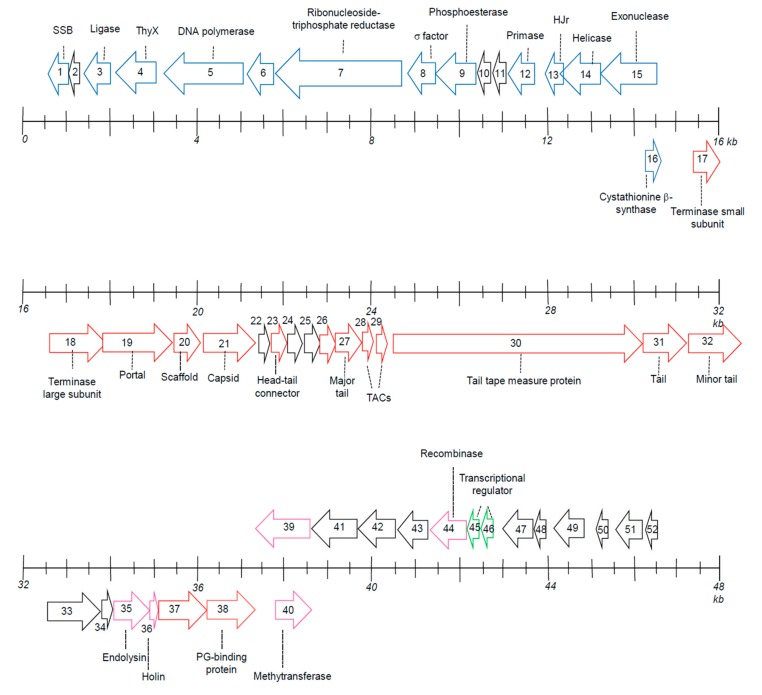
Organization of ORFs in the P318 genome. The scales represent the length of the genome and the nucleotide positions in kb are given. Arrowed boxes denote the predicted ORFs in the genome. ORFs pointing to the right are on the forward strand, while those pointing to the left are on the complementary strand. ORFs colored blue, red, green, and magenta represent proteins involved in phage DNA replication, virion morphogenesis, transcription control, cell wall lysis, DNA modification, and DNA recombination. The abbreviations are as follow: SSB, ssDNA binding protein; ThyX, FAD-dependent thymidylate synthase; HJr, Holliday junction resolvases; TACs, tail assembly chaperone; PG, peptidoglycanwhich may protect phage DNA from host restriction endonuclease [26].

**Figure 6 viruses-11-01042-f006:**
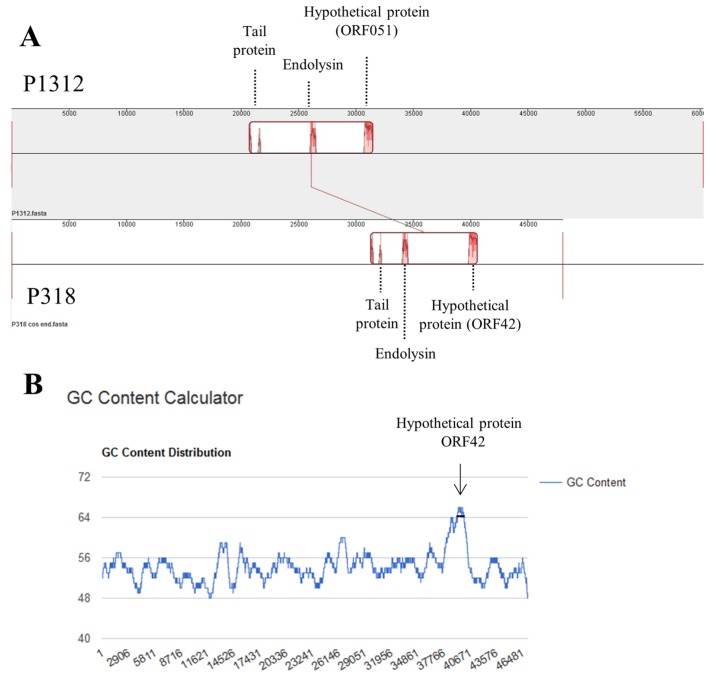
Genome comparison between bacteriophages P318 and P1312. (**A**) The similarity in nucleotide sequences of the two phages was analyzed by the Mauve alignment. The regions with similarity are boxed and the height of signal reflects the similarity level. (**B**) The G+C content is calculated along the P318 genome with a window size of 1000 base pairs by the method provided in the website https://www.biologicscorp.com/tools/GCContent/.

**Table 1 viruses-11-01042-t001:** General features of the proteins encoded by the predicted ORFs of the P318 genome.

ORF	Start (nt)	End (nt)	Size (kDa) ^a^	Predicted Functions	BLASTP (Best Match)	LC-M/M ^b^
1	1255	827	15.8	ssDNA binding protein	YP_009269232.1 *Gordonia* phage Smoothie	
2	1461	1240	8.2	hypothetical protein	WP_015060945.1 *Streptomyces* sp.	
3	2142	1534	22.4	RNA ligase	YP_008859155.1 *Mycobacterium* phage Zaka	
4	3163	2312	31.3	FAD-dependent thymidylate synthase	WP_043914404.1 *Kitasatospora griseola*	15
5	5103	3340	64.7	DNA polymerase	OEV28665.1 *Streptomyces nanshensis*	
6	5717	5181	19.7	phage protein	SBO90589.1 *Nonomuraea gerenzanensis*	
7	8719	5762	109.6	ribonucleoside-triphosphate reductase	OEV28666.1 *Streptomyces nanshensis*	4
8	9570	8872	25.7	putative sigma factor	AIS73726.1 *Mycobacterium* phage Quinnkiro	
9	10365	9580	28.4	phosphoesterase	YP_008051820.1 *Mycobacterium* phage Severus	29
10	10676	10362	11.6	hypothetical protein	OEV28668.1 *Streptomyces nanshensis*	
11	10950	10669	10.5 (V)	hypothetical protein	WP_093886506.1 *Streptosporangium canum*	
12	11769	11149	22.8 (V)	DNA primase	ASZ75224.1 *Mycobacterium* phage MissWhite	
13	12347	11955	14.5	resolvase	AYN59122.1 *Arthrobacter* phage Yang	
14	13164	12322	31.0	DnaB-like helicase	AFF28358.1 *Mycobacterium* phage Twister	
15	14397	13441	35.1	exonuclease	AOT25428.1 *Mycobacterium* phage BabyRay	
16	15206	15394	7.0 (V)	cystathionine β-synthase	WP_087442904.1 *Paenibacillus thiaminolyticus*	
17	15851	16300	16.5	terminase small subunit	AYN58482.1 *Arthrobacter* phage Maureen	
18	16740	17888	42.2	terminase large subunit	SFI82078.1 *Streptosporangium canum*	
19	17902	19350	54.1	phage portal protein	WP_093886565.1 *Streptosporangium canum*	17
20	19402	20007	22.3	phage scaffold protein	AWN05292.1 *Streptomyces* phage Ibantik	7
21	20023	21084	39.0	phage major capsid protein	AWN05293.1 *Streptomyces* phage Ibantik	47
22	21166	21432	7.8	hypothetical protein	OEV28690.1 *Streptomyces nanshensis*	18
23	21435	21800	13.4	phage gp6-like head-tail connector protein	WP_018564996.1 *Streptomyces* sp.	9
24	21797	22111	11.6	hypothetical protein	WP_055523439.1 *Streptomyces graminilatus*	25
25	22147	22557	15.1	hypothetical protein	WP_039630012.1 *Streptomyces*	14
26	22557	22937	11.4	DUF3168 domain-containing protein	WP_120720997.1 *Streptomyces hundungensis*	27
27	22941	23573	23.3	phage major tail protein	SBO90565.1 *Nonomuraea gerenzanensis*	31
28	23667	23963	10.9	tail assembly chaperone	YP_009302776.1 *Gordonia* phage BetterKatz	
29	23978	24397	15.4	tail assembly chaperone	AXH45781.1 *Gordonia* phage GEazy	
30	24417	30116	207.5	tape measure domain -containing protein	SMB97758.1 *Thermanaeromonas toyohensis*	24
31	30128	31168	38.2	phage tail protein	WP_079249051.1 *Streptomyces* sp.	8
32	31168	32589	52.2	phage tail minor protein	ASM62305.1 *Arthrobacter* phage Nightmare	21
33	32582	33700	41.1	hypothetical protein	WP_033100528.1 *Thermoactinomyces daqus*	34
34	33717	34019	11.2 (V)	hypothetical protein	WP_004938179.1 *Streptomyces mobaraensis*	
35	34016	34828	29.9	N-acetylmuramoyl-L-alanine amidase	ALA06408.1 *Thermobifida* phage P1312	
36	34880	35098	8.1	hypothetical protein	WP_026415462.1 *Actinomadura oligospora*	15
37	35098	36045	34.8	phage tail protein	WP_079733063.1 *Acinetobacter baumannii*	25
38	36045	36992	34.8	Peptidoglycan binding domain containing protein	WP_013490242.1 *Bacillus cellulosilyticus*	5
39	38528	37050	54.3	DUF3987 domain-containing protein	WP_012850795.1 *Thermomonospora curvata*	13
40	37395	38753	49.9	putative methyltransferase	EUA88088.1 *Mycobacterium ulcerans* str. Harvey	
41	39792	38629	42.7	hypothetical protein	ONK09418.1 *Streptomyces* sp. MP131-18	
42	40601	39894	28.9	hypothetical protein P1312_051	ALA06416.1 *Thermobifida* phage P1312	8
43	41401	40742	23.6	hypothetical protein	OEV28683.1 *Streptomyces nanshensis*	
44	42099	41398	25.8 (V)	recombinase	WP_098165210.1 *Bacillus pseudomycoides*	
45	42541	42224	11.7	MerR family transcriptional regulator	WP_003947861.1 *Streptomyces*	
46	42863	42630	8.6	peptidoglycan editing factor PgeF	WP_069300862.1 *Neptunicoccus sediminis*	
47	43590	43108	17.8	hypothetical protein	XP_005789494.1 *Emiliania huxleyi* CCMP1516	4
48	43997	43731	9.8 (V)	hypothetical protein	WP_060729217.1 *Streptomyces albus*	
49	44740	44102	23.5	No match		
50	45599	45249	12.9	hypothetical protein	WP_071806278.1 *Couchioplanes caeruleus*	
51	46086	45673	15.2	No match		
52	46447	46124	11.9	DUF305 domain-containing protein	WP_009082635.1 *Actinobacteria*	

a: Proteins with Valine as their first translated amino acid is indicated with (V); b: The coverage (%) of amino acid sequence determined by mass spectrometry.

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
