# Peer review of "Characterization of a Novel Thermobifida fusca Bacteriophage P318"

_viruses, 2019, doi:10.3390/v11111042_

Round 1

Reviewer 1 Report

This is a nice paper.

In the 4th line of the Abstract "learnt" should be "learned."

In the 9th line of the Abstract "and others" should be replaced with more detail.

On page 2 line 53, "lysogenic" should be changed to "temperate." See Andre Lwoff, Lysogeny, Bacteriological Reviews 17:269-337, 1953.

On page 9, line 216, the meaning of "hind" is not clear to me.

Author Response

Reviewer #1
We are grateful to this reviewer for his/her insightful comments on our paper. We have been able to incorporate changes to reflect most of the suggestions provided by the reviewer. We have highlighted the changes within the manuscript.
1. In the 4th line of the Abstract "learnt" should be "learned."
The sentence was rephrased as shown in Lines 10-11.
2. In the 9th line of the Abstract "and others" should be replaced with more detail.
“and others” was changed to “and the regulation of the lytic/lysogenic cycle”. Line15
3. On page 2 line 53, "lysogenic" should be changed to "temperate."
Change was made. Line 49
4. On page 9, line 216, the meaning of "hind" is not clear to me.
It was changed to “rear”. Line 222

Reviewer 2 Report

The manuscript ny Cheepudom et al. presents a fairly standard description of a novel Siphoviridae phage against Thermobifida fusca. The manuscript is well written and laid out in a logical order. The experiments were properly designed and mostly correctly interpreted.

General remark:

While the description of this new phage is completed to a satisfactory level, an important piece of information on the phage lifestyle is missing. Authors correctly point out, that bacteriophages (especially temperate ones) could be used as tools for manipulating this industrially-important bacterium. Therefore, doing a few extra experiments to confirm temperate lifestyle and identify the site for integration into bacterial chromosome are absolutely essential. The manuscript can also benefit from shortening and removal of un-necessary repetitions (e.g. between the Methods and the Results sections) and lengthy and very basic explanations.

Specific remarks:

Lines 50-53: Given the audience of this journal, such a basic introduction into utility of bacteriophages is not needed.

Lines 88-89: In this size-exclusion chromatography setup, do virions, due to their large size, simply elute with the void volume of the column?

Lines 142-146: Use 'restriction fragments' and 'restriction patterns' instead of 'restricted bands' and 'restricted patterns'.

Lines 176-177: The latent period of just 10 min seems to be very short in comparison with T. fusca doubling time (4 hours if I'm not mistaken). Could it be that some steps of phage replication were already taking place during wash steps (described with line 95)? I wonder if you could repeat the experiment omitting the washes? Since you infect at low MOI most of the phage should be bound to cells anyway and the background concentration will be very low. This might give you a more accurate estimate of latent period.

Lines 180-182: This sentence repeats what was already said in the Methods section.

Lines 183-191: This is too basic and explanatory for the audience of this journal. Lines 193-195 will be sufficient.

Lines 196-200, 203-213: Again too explanatory and also overlaps with methods section. This whole paragraph can be reduced to 2-3 sentences.

Figure 4 and Table 1 are useful, but not absolutely essential to the reader and can be moved to supplementary data.

Lines 249-253, 281-285, 299-302: Un-necessary explanations...

Lines 247-307: This whole section should be shortened. Also, I suggest avoiding the use of language that implies that functions of the named protein products are fully established. In reality, you only report homology to proteins with established functions and hypothetically assign functions to proteins in P318 with certain level of confidence. Please bear that in mind and be less categorical with P318 proteins function claims.

Lines 308-315: This last gene module bears similarity to classical lysogeny modules. I think it is absolutely essential to experimentally test lysogenic potential of this phage.

Figure 6 should be moved to supplementary.

Author Response

Reviewer 2
1. Lines 50-53: Given the audience of this journal, such a basic introduction into utility of bacteriophages is not needed.
# The original sentence was deleted.
2. Lines 88-89: In this size-exclusion chromatography setup, do virions, due to their large size, simply elute with the void volume of the column?
# The sentence was rephrased to include the additional information. Lines 85-86
3. Lines 142-146: Use 'restriction fragments' and 'restriction patterns' instead of 'restricted bands' and 'restricted patterns'.
# Changes were made throughout the text regarding this comment.
4. The latent period of just 10 min seems to be very short in comparison with T. fusca doubling time (4 hours if I'm not mistaken). Could it be that some steps of phage replication were already taking place during wash steps (described with line 95)?....
# The wash step was carried out at 4oC to minimize the replication of P318 that had invaded the host cell so that the latent period time could be estimated as accurate as possible. This operation condition was added in Line 92.
5. Lines 180-182: This sentence repeats what was already said in the Methods section.
# The redundant sentences were deleted and the revised sentence is shown in Line 185-186.
6. Lines 183-191: This is too basic and explanatory for the audience of this journal. Lines 193-195 will be sufficient.
# The reviewer has raised an important point here. However, we believe that this information would be more appropriate for the general readers who do not have specific knowledges about the packaging mechanisms of bacteriophages. (Line 190-195).
7. Lines 196-200, 203-213: Again too explanatory and also overlaps with methods section. This whole paragraph can be reduced to 2-3 sentences.
# The description about how the P318 genome was treated by BAL-31 and restriction enzymes was now mainly written in Materials and Methods. Only a short explanation is given in the Results and Discussion section to enable the readers more easily to understand this work.
# We believe that the paragraph (Lines 196-225 in the original version) that describes how the distal ends of the P318 genome was actually determined is an important part of this work. Therefore, we would like to keep it in the text.
8. Figure 4 and Table 1 are useful, but not absolutely essential to the reader and can be moved to supplementary data. Figure 6 should be moved to supplementary.
# Having Fig. 4, Fig. 6 and Table 1 kept in the main text may be a more appropriate setting because this format may help the readers to understand this work more easily.
9. Lines 249-253, 281-285, 299-302: Un-necessary explanations...
# In response to this comment, we have deleted these unnecessary descriptions.
10. Lines 247-307: This whole section should be shortened.
# The paragraphs were reconstructed so that they appear more concise (Line257-302) to conform to the idea of the reviewer’s comments.
11. Lines 308-315: This last gene module bears similarity to classical lysogeny modules. I think it is absolutely essential to experimentally test lysogenic potential of this phage.
# We agree with the reviewer on this comment. Nonetheless, it will take time and labor to confirm the temperate lifestyle of P318 and identify the site for integration into bacterial chromosome. This will be investigated in our future study.

Reviewer 3 Report

The manuscript "Characterization of a novel Thermobifida fusca bacteriophage P318" describes the isolation of a novel thermostable bacteriophage, isolated from the soil sample. The bacteriophage was fully characterized, including genome analysis, structural proteome, and biological properties. Apparently, the phage is unique, as the genome nucleotide sequence has no detectable similarity with other bacteriophage genomes, except a low similarity with another Thermobifida fusca bacteriophage P1312. 

Considering the novelty of the phage, I recommend the manuscript to publication after major revision.

Major Essential Revisions:

1. The results and discussion section does not quite correspond to the materials and methods section. There is no description of phylogenetic analysis and software used for phylogenetic analysis in the Materials and Methods section. The same concerns to protein gel electrophoresis and mass spectrometry analysis, which are not described in the methods.

2. Page 4, Line 124, 125. Missing primer sequences or references to them. Figure 1B. It is desirable to sign protein bands corresponding to the identified by MS/MS analysis phage proteins.

3. Figure 3. Please, add GenBank accession numbers to the terminase large subunits of different phages, which were used for phylogenetic analysis.

The authors classify the phage P318 according to its virion morphology. Is it possible to clarify the taxonomy of this phage, using phylogenetic analysis of DNA polymerase, capsid protein or other proteins?

4. Page 9, Line 216, Missing primer sequence.

5. Chapter "Genome structure of P318" in the Results and Discussion: Please, add the data, concerning genome fragment B. Please, explain, why this fragment is not represented in Figure 4A.

6. Three nucleotide metabolic proteins, encoded with ORF4, ORF7, and ORF9, were identified in purified virions using MS/MS (Table 1). Can authors suppose their function during phage infection and any mechanisms which are used by these proteins to penetrate into the bacterial cell?

7. Page 7, line 197: Enzymes XhoI and Hind III do not cut RNA, why do the authors call them "restriction endoribonucleases"? Please, add the abbreviation for the term "BAL-31", which you use instead of "BAL-31 exonuclease".

8. The manuscript needs adequate professional proofreading, as it contains a number of terms that cause misunderstanding, e.g.:

Page 3, line 120: "...HindIII-restricted..." 

Page 9, line 204: "...the pretended circular map..."

Page 15, line 292: "tape measurement protein..."

Author Response

Reviewer 3
1. The results and discussion section does not quite correspond to the materials and methods section. There is no description of phylogenetic analysis and software used for phylogenetic analysis in the Materials and Methods section. The same concerns to protein gel electrophoresis and mass spectrometry analysis, which are not described in the methods.
# The methodology of phylogenetic analysis and LC-MS/MS is added in this revision. Lines 96-101 and Lines 128-135
2. Page 4, Line 124, 125. Missing primer sequences or references to them. Figure 1B. It is desirable to sign protein bands corresponding to the identified by MS/MS analysis phage proteins.
# The sequences of M13 primers are given in Lines 124-125 in this revision.
# Only the virions purified via the gel filtration chromatography was sent for protein identification by mass spectrometry. We did not analyze the individual protein band that appeared on the SDS-PAGE gel. Although we can guess the identity of the band based on its estimated molecular mass, it is still speculative. Therefore, we decide not to put the labels along the protein bands.
3. Figure 3. Please, add GenBank accession numbers to the terminase large subunits of different phages, which were used for phylogenetic analysis.
# The accession number of each of the terminase large subunits is added in Fig. 3.
The authors classify the phage P318 according to its virion morphology. Is it possible to clarify the taxonomy of this phage, using phylogenetic analysis of DNA polymerase, capsid protein or other proteins?
# The classification of a tailed-bacteriophage within the Caudovirales depends on its morphology. Phylogenetic analysis on the terminase large subunit is to predict the packaging mechanism of the tailed-bacteriophages.
4. Page 9, Line 216, Missing primer sequence.
# The cosF’ primer sequence was added in line 126.
5. Chapter "Genome structure of P318" in the Results and Discussion: Please, add the data, concerning genome fragment B. Please, explain, why this fragment is not represented in Figure 4A.
# The label of fragment B is added in Fig. 4A. It is indistinguishable in this gel due to its size (2.7 kb) is close to the sizes of the neighboring bands (fragment 5, 2.8 kb). The description of fragment B is added in Lines 207-209.
6. Three nucleotide metabolic proteins, encoded with ORF4, ORF7, and ORF9, were identified in purified virions using MS/MS (Table 1). Can authors suppose their function during phage infection and any mechanisms which are used by these proteins to penetrate into the bacterial cell?
# The presence of the three nucleotide metabolic enzymes with virions is bewildering. It implies that they either play an unidentified role in the invasion process of P318 or interact nonspecifically with the viral structural proteins. These sentences are given in Lines 251-253.
7. Page 7, line 197: Enzymes XhoI and HindIII do not cut RNA, why do the authors call them "restriction endoribonucleases"? Please, add the abbreviation for the term "BAL-31", which you use instead of "BAL-31 exonuclease".
# The word “restriction endoribonucleases” was deleted throughout the text. The abbreviation of BAL-31 exonuclease as "BAL-31" is added in line 115.
8. The manuscript needs adequate professional proofreading, as it contains a number of terms that cause misunderstanding,
# To conform to the reviewer suggestion, this revised manuscript was edited by a native English-speaking scholar.
9. Page 3, line 120: "...HindIII-restricted..."
#It was changed to HindIII-digested. Line 120
10. line 204: "...the pretended circular map..."
# The word “pretended” was deleted and the sentence was rewritten. We believe that these revisions give greater clarity to the points being made. Line 210-211
11. Page 15, line 292: "tape measurement protein..."
# “measurement” was changed to “measure”. Line 289

Round 2

Reviewer 1 Report

Nice paper.

Author Response

No comment needs to be answered.

Reviewer 2 Report

The authors have completely dismissed my key point about determining the phage lifestyle.

I think from a study like this, the first thing a reader would like to know is whether the newly described phage is virulent or temperate. This has never been demonstrated or even speculated about in the manuscript. Although the efficiency of lysogeny experiment and confirmation of phage integration into chromosome will take a bit of time, this additional work is neither hugely time consuming, nor very expensive to do.

My opinion here remains unchanged and I strongly suggest to the editor that these additional experiments should be performed and reported in the revised version of this manuscript.

Lines 293-294: "The lysogeny cassette of P318 includes the N-acetylmuramoyl-L-alanine amidase (ORF35), the holin protein (ORF36), and a peptidoglycan (PG)-binding domain (ORF38)" - endolysin (amidase) and holin are NOT part of lysogeny module! They are located in oppositely oriented operon, together with structural proteins genes and constitute a lytic gene module. You can see that from Fig. 5.

Author Response

The suggested work regarding the integration site of P318 is beyond the scope of the current study. Nonetheless, the concern is addressed in Results and Discussion in Lines 309-311.  The name of the module composed of endolysin (ORF35), holin (ORF36) and PG-binding domain (ORF38) was changed from "lysogeny module" to "lytic gene module". Line 292. 

Reviewer 3 Report

Dear Authors,

In general, I was satisfied with the improvements made to the article and the authors ' answers. The only point I would like to repeat is that it would be interesting to more accurately determine the taxonomic position of the phage within the family Siphoviridae based on phylogenetic analysis of bacteriophage protein sequences. However, this is not an extremely important point in this study and the article can be accepted for publication in its present form.

Author Response

To further determine the taxonomic position of P318 within the family Siphoviridae, the phage major capsid protein was used as the marker in a phylogenetic tree analysis, in which 166 phage proteins from 157 genera of the Siphoviridae family were included. Nonetheless, the result did not conclusively indicate the taxonomic position of P318. Therefore, the taxonomy of P318 within the Siphoviridae family is uncertain at this moment.